# TMDiMP: Temporal Memory Guided Discriminative Tracker for UAV Object Tracking

**Zheng Yang** [1], **Bing Han** [1,*], **Weiming Chen** [1] and **Xinbo Gao** [1,2]

1   School of Electronic Engineering, Xidian University, Xi'an 710071, China
2   Chongqing Key Laboratory of Image Cognition, Chongqing University of Posts and Telecommunications, Chongqing 400065, China
*   Correspondence: bhan@xidian.edu.cn

**Abstract:** Unmanned aerial vehicles (UAVs) have attracted increasing attention in recent years because of their broad range of applications in city security, military reconnaissance, disaster rescue, and so on. As one of the critical algorithms in the field of artificial intelligence, object tracking greatly improves the working efficiency of UAVs. However, unmanned aerial vehicle (UAV) object tracking still faces many challenges. UAV objects provide limited textures and contours for feature extraction due to their small sizes. Moreover, to capture objects continuously, a UAV camera must constantly move with the object. The above two reasons are usual causes of object-tracking failures. To this end, we propose an end-to-end discriminative tracker called TMDiMP. Inspired by the self-attention mechanism in Transformer, a novel memory-aware attention mechanism is embedded into TMDiMP, which can generate discriminative features of small objects and overcome the object-forgetting problem after camera motion. We also build a UAV object-tracking dataset with various object categories and attributes, named VIPUOTB, which consists of many video sequences collected in urban scenes. Our VIPUOTB is different from other existing datasets in terms of object size, camera motion speed, location distribution, etc. TMDiMP achieves competitive results on our VIPUOTB dataset and three public datasets, UAVDT, UAV123, and VisDrone, compared with state-of-the-art methods, thus demonstrating the effectiveness and robustness of our proposed algorithm.

**Keywords:** unmanned aerial vehicle; object tracking; small object; camera motion

## 1. Introduction

Unmanned aerial vehicles (UAVs) have played an important role in aviation remote sensing because of their flexibility and safety. With the development of artificial intelligence in recent years, UAVs have been widely used in many computer vision areas, such as intelligent surveillance [1], disaster rescue [2], smart agriculture [3], city security [4], and so on. Among these applications, object tracking is one of the most fundamental and important algorithms.

Object tracking can be divided into single-object tracking (SOT) and multiobject tracking (MOT). SOT aims to estimate the location of a certain object in a video sequence. More studies have proposed many visual tracking methods, such as discriminative trackers [5–9] and Siamese-based trackers [10–14]. All of these effective models mainly focus on object tracking in surveillance scenarios. Different from generic object tracking, unmanned aerial vehicle (UAV) object tracking follows objects from an aerial perspective. According to research on UAV object-tracking data, we find that there are two main differences between UAV object tracking and generic object tracking.

The first challenge is the small size of objects in UAV scenes. The relative distances between the UAV and objects are larger than those of other sensors and objects. Most of the objects in video sequences captured by UAV occupy less than 1% of the total number of pixels. The information we can obtain from these small objects is limited, especially the

appearance features to represent the objects. Furthermore, appearance features of small objects are easily influenced by backgrounds. Most existing methods for object tracking cannot track objects successfully by using limited features in complex UAV scenes.

The second challenge usually occurs during the drone flight. To capture objects continuously, a camera mounted on a UAV will move with the tracked object frequently. According to our observation, trackers usually lose the object after the UAV camera moves suddenly, which we call the target-forgetting problem.

In fact, the smallness of objects and camera motion are two common attributes in the aviation remote sensing field. For example, urban traffic surveillance has the characteristics of high spatial complexity [15]. In order to obtain more information about traffic scenes, the altitudes of UAV views are usually high, which leads to small sizes of objects. In addition, it is necessary to adjust shooting angles of the onboard cameras; otherwise, the UAV will lose the tracked objects that move very quickly.

Many research works have shown their effectiveness for UAV object tracking [16–26], and some of them have been concerned about the problems mentioned above. To represent small objects, Wang [18] employed locally adaptive regression kernel (LARK) features to encode the edge information of the objects. However, the edges of objects are sometimes fuzzy when the background is complex or the object is small and blurred. Li [17] proposed a geometric transformation based on background feature points for camera motion. However, this method did not utilize the deep features of CNN-based trackers. The problems caused by small objects and camera motion are summarized as difficult attributes in the UAV database proposed by Du [27].

Through the experiment results and observation, we find that temporal information in sequences can enhance features of small objects and overcome the target-forgetting problem. Therefore, in order to cope with target-missing problems caused by insufficient representation of small objects and the simultaneous sudden movement of cameras, we design a memory-aware attention mechanism to leverage temporal memory in video clips. There are already many existing trackers [28–30] that have utilized temporal information to boost tracking performance. However, these trackers extracted temporal information from high-level features of images and ignored temporal information contained in low-level features; however, low-level features are useful for object tracking because they contain location and boundary information [9]. Thus, the proposed memory-aware attention mechanism can utilize temporal information contained in low-level features to improve the feature representation ability of trackers, and also encourage trackers to learn the pattern of camera motion in UAV object tracking. We embed the memory-aware attention mechanism into a discriminative tracker discriminative model prediction (DiMP) [7], resulting in a new framework, named the temporal memory-guided DiMP (TMDiMP).

In addition, there is a lack of benchmark datasets devoted to visual tracking, especially containing small objects and camera motion, which are ubiquitous in real-world scenes. Many features, small objects, and camera motion are the usual reasons for object-tracking failure. Thus, to solve the problem caused by small objects and camera motion, we build a UAV object-tracking benchmark named VIPUOTB. We also define the average proportion of the target size to an image (APTS) and the average moving distance between adjacent frames during camera motion (AMDAF) to measure the normalized object size and camera motion speed. In VIPUOTB, the APTS is the smallest compared with all generic object tracking datasets [31–33] and other UAV object-tracking datasets [17,27,34–36]. The AMDAF is also the largest compared with other UAV object-tracking datasets. The VIPUOTB is used to verify that our TMDiMP can cope with the tracking problems caused by small objects and camera motion.

In particular, the research scope of our study is aerial remote-sensing image processing based on advanced artificial intelligence technology. Specifically, we focus on designing a powerful UAV object-tracking method to mitigate the issues caused by small objects and camera motion. During our study, we also generate a new UAV object-tracking dataset that contains smaller objects and faster camera motion speed compared with other datasets. The main contributions of our work are summarized as follows.

- We present a specially designed framework with end-to-end training capabilities, called TMDiMP, which embeds a novel memory-aware attention mechanism. Temporal memory is utilized to generate discriminative features of small objects and overcome the object-forgetting problem of camera motion; thereby, the tracker can obtain more accurate results in complex UAV scenes.
- We build a UAV object-tracking dataset collected in urban scenes, named VIPUOTB, which contains various object categories and different attributes. All video sequences in our dataset are labeled manually by several experts to avoid subjective factors. Compared with other existing UAV datasets, VIPUOTB is different in terms of object size, camera motion speed, location distribution, etc.
- The quantitative and perceptive experimental results illustrate that our proposed TMDiMP achieves competitive performance compared with state-of-the-art methods on our VIPUOTB dataset and three public datasets, UAVDT, UAV123, and VisDrone.

The structure of this paper is as follows: other works related to ours are reviewed in Section 2. A description of the proposed TMDiMP is given in Section 3. The generated dataset VIPUOTB is introduced in Section 4. The experimental results obtained by our method are presented in Section 5. A discussion and conclusions from this work are presented in Sections 6 and 7.

## 2. Related Works

### 2.1. The Discriminative Online Learning Trackers

Object-tracking methods can be roughly divided into discriminative trackers and Siamese-based trackers [7]. The Siamese trackers have demonstrated their end-to-end training capabilities and competitive performance in tracking accuracy. Compared with Siamese trackers, discriminative online learning trackers can effectively utilize the information from background regions and previous tracking frames.

Because this paper mainly focuses on small-object tracking, it is necessary to utilize background information and temporal information for more accurate and robust tracking. Therefore, we choose the discriminative online learning tracker as our baseline.

The discriminative trackers can be further divided into correlation filter-based trackers [5,6,37–39] and CNN-based trackers [7–9,28–30,40,41]. CNN-based trackers have gained more attention in recent years due to their outstanding image-representation power. MDNet [9] first learned shared representation of targets from multiple videos, then MDNet regarded each video as a separate domain and learned domain-specific information through online learning during tracking. Vital [41] utilized adversarial learning to augment positive samples in the feature space and captured rich appearance variations of targets. Vital also handled the issue of class imbalance in the training stage by using the proposed higher-order cost sensitive loss. ECO [40] was proposed to simultaneously improve both the correlation filter-based trackers' speed and robustness by using a novel factorized convolution operator and an efficient model update strategy. STRCF [39] leveraged temporal regularization to provide a robust appearance model for object tracking and handled the problem of boundary effects contained in correlation filter-based trackers. ATOM [42] improved the accuracy of target sizes prediction by introducing a novel estimation component, which can learn high-level knowledge through extensive offline training. Inspired by [9,42], DiMP [7] was carefully designed to maximize the discriminative ability of the predicted model. In DiMP, a discriminative learning loss and a powerful optimization strategy were used to promote the robustness of the predicted mode and ensure rapid convergence, respectively. PrDiMP [8] modified DiMP by predicting targets' conditional

probability densities, which allowed the computation of absolute probabilities. However, the accuracy of these methods will decrease when they are directly used for small-object tracking. Therefore, we design a novel framework TMDiMP, which improves the discriminative tracker by temporal context propagation, to solve the problems caused by small objects and camera motion simultaneously.

### 2.2. UAV Object Trackers

In this section, we mainly introduce UAV object-tracking methods, which are usually proposed by modifying classic trackers. Fang [16] modified the mean shift algorithm to track the objects with fast motion in UAV videos; Bai [23] proposed an attention-based mask generative network to handle the problems of occlusion and deformation. Li [17] designed an algorithm to solve the problem of camera motion estimation. Sun [25] proposed a template-driven Siamese network that can adapt well to frequent appearance change in UAV video datasets. Wang [18] proposed an appearance model based on the locally adaptive regression kernel for small UAV object tracking by encoding the geometric structure of the objects. Zhang [19] proposed a coarse-to-fine deep scheme by modifying the ADNet [43] to address the problem of aspect ratio changing. The coarse tracker and fine tracker have their own action space and operator. Similar to [19], Song [20] proposed a boundary-decision network for the ARC problem. For long-term tracking, Li [21] proposed a tracker named FAST, which exploits the inherent correlation between the frequency tracker and spatial detector. In consideration of the limitation of computational resources onboard UAV, Li [22] designed a correlation filter-based method with high efficiency, called Autotrack, which could automatically and adaptively learn spatiotemporal regularization terms to improve the learning of objects. Our TMDiMP can not only address small objects and camera motion, which are considered the main reasons for the poor performance in the field of UAV tracking, but also cope with most of the abovementioned attributes.

### 2.3. Trackers Using Temporal Information

Temporal information analysis is critical to success in many fields of video understanding and analysis, such as action recognition [44], trajectory prediction [45] and video retrieval [46]. Many previous studies also introduced temporal information into object tracking. Teng [47] incorporated temporal and spatial information to boost tracking performance by a deep architecture with three subnetworks: a feature network, a temporal network and a spatial network. Gao [48] presented a spatiotemporal graph convolutional network method for visual tracking. KYS [28] was proposed by modifying DiMP, which could utilize a dense set of localized state vectors to represent scene information and achieve an improved scene-aware target prediction in each frame. TrDiMP [29] introduced a Transformer architecture into DiMP to explore the temporal contexts across video frames. STM [30] used historical information of the target by a spacetime memory network for better adapting to appearance variations during tracking. At last, Table 1 is presented to summarize contributions and limitations of some most recent relevant methods. Compared with these existing methods, our proposed memory-aware attention mechanism can utilize temporal information contained in low-level features to achieve better representation ability.

**Table 1.** Contributions and limitations of some most recent relevant methods.

| Method | Contributions | Limitations |
| --- | --- | --- |
| 2020' PrDiMP [8] | It predicted the conditional probability density of targets. | It did not utilize the temporal information of video sequences. |
| 2020' KYS [28] | It leveraged change information of targets' surroundings between adjacent frames. | It ignored temporal information contained in low-level features. |
| 2021' TrDiMP [29] | It extracted temporal contexts among frames by using a Transformer architecture. | It ignored temporal information contained in low-level features. |
| 2021' STM [30] | It leveraged historical information of targets by using a novel module. | It ignored temporal information contained in low-level features. |
| 2021' Autotrack [22] | It was capable of efficient tracking with low computational requirements. | It did not utilize the temporal information of video sequences. |
| 2022' TDsiam [25] | It leveraged historical information of targets by using a novel module. | It ignored temporal information contained in low-level features. |
| 2022' AMGN [23] | It proposed an mask generation network to handle problems caused by occlusion and deformation. | It did not utilize the temporal information of video sequences. |

### 2.4. UAV Object-Tracking Dataset

VIVID [36] is a UAV-based dataset including only nine sequences, proposed by Collins. Li [17] proposed a dataset DTB70 built on a university campus. UAV123 [34] consists of 123 sequences created for UAV-based tracking. Various scenarios and tracking objects exist in UAV123. The attributes of the tracking problem include background clutter, occlusion, illumination variation, camera motion, viewpoint change, scale variation and so on. UAVDT [27] is a UAV dataset not only for single-object tracking but also for multiobject tracking. The data for single-object tracking includes 50 sequences with only cars, trucks and buses. The video attributes in UAVDT are similar to those in UAV123. VisDrone [35] is by far the largest dataset with 167 sequences for single-object tracking, which is divided into training, validation, and testing sets. Multiobject tracking is also considered in VisDrone. We construct a new UAV object tracking dataset, VIPUOTB, which is different from other existing datasets in terms of object size, camera motion speed, location distribution, etc.

## 3. Proposed Framework

In this section, we first introduce the memory-aware attention mechanism. Next, the proposed TMDiMP framework is demonstrated, and the details of the network are outlined.

### 3.1. The Memory-Aware Attention Mechanism

The attention mechanism is one of the key components, which is widely used in different fields, such as feature representation [49] and network architecture [50], especially in Transformer, proposed by Vaswani [51] for natural language processing. An attention function can be described as mapping a query and a set of key-value pairs to an output, where the query, keys, values, and output are all vectors [51].

The self-attention mechanism proposed in Transformer can automatically focus on the interesting region of an image, which contains more useful information for the constructed task. Inspired by self-attention, we propose a memory-aware attention mechanism to generate discriminative feature maps of small objects and overcome the problem of object forgetting in trackers.

The low-level feature of targets contains the location and boundary information, which is beneficial for object tracking [9]. In addition, extracting low-level features will not result in too much computation in the online tracking stage. Therefore, we first extract the low-level features $F_t^{LL}$ and $F_{t-1}^{LL}$ of a current frame $I_t$ and a previous frame $I_{t-1}$.

Then, $F_t^{LL}$ and $F_{t-1}^{LL}$ are transformed into two feature spaces $h$ and $g$ to obtain key $K$ and query $Q$, respectively,

$$\begin{cases} K = h(F_t^{LL}) \\ Q = g(F_{t-1}^{LL}) \end{cases} \tag{1}$$

A batchwise matrix multiplication and a *Softmax* layer are then applied, resulting in a corresponding attention map $M_{t,t-1}$, which contains the temporal memory

$$M_{t,t-1} = Softmax(Q \otimes K), \tag{2}$$

where $M_{t,t-1}$ indicates the attention of all points on key $K$ for each point on query $Q$.

In classic self-attention [51], the corresponding attention map $M_{t,t-1}$ is then multiplied by the value $V$ and added back to $V$ for direct feature enhancement. The $V$ is usually obtained by transforming $F_t^{LL}$ by using another simple convolutional layer. The memory feature $F_t^{me}$, which is output of the attention mechanism, can be calculated by Equation (3),

$$F_t^{me} = M_{t,t-1} \otimes (V) \oplus (V), \tag{3}$$

where $V = conv(F_t^{LL})$, and *conv* is a convolutional layer.

As described above, to obtain location and boundary information, we use the low-level features to compute the corresponding attention map $M_{t,t-1}$ in Equation (2). While acting as our baseline, discriminative trackers usually utilize the semantic information to distinguish the target object from the background, and low-level features contain less semantic information, which is contained in high-level features. Therefore, we use the high-level features as the value $V$; thereby, the memory-aware attention mechanism can merge $M_{t,t-1}$ and high-level features in attention processing.

Due to the different sizes of low-level features and high-level features, we need to downsample $M_{t,t-1}$ to the same size as $V$. Then, an enhancing layer is employed to merge $M_{t,t-1}$ and $V$. Thus, the Equation (3) is modified as Equation (4) to calculate $F_t^{me}$. We have

$$F_t^{me} = Enh(Ds(M_{t,t-1}), V), \tag{4}$$

where $V = Bb(F_t^{LL})$. *Enh*, *Ds* and *Bb* denote the enhancing layer, downsampling layer, and backbone, respectively.

### 3.2. Overall View of TMDiMP

As shown in Figure 1, our proposed TMDiMP comprises three components, a backbone, a memory-aware attention module, and two prediction branches. The workflow of our method is represented in Algorithm 1.

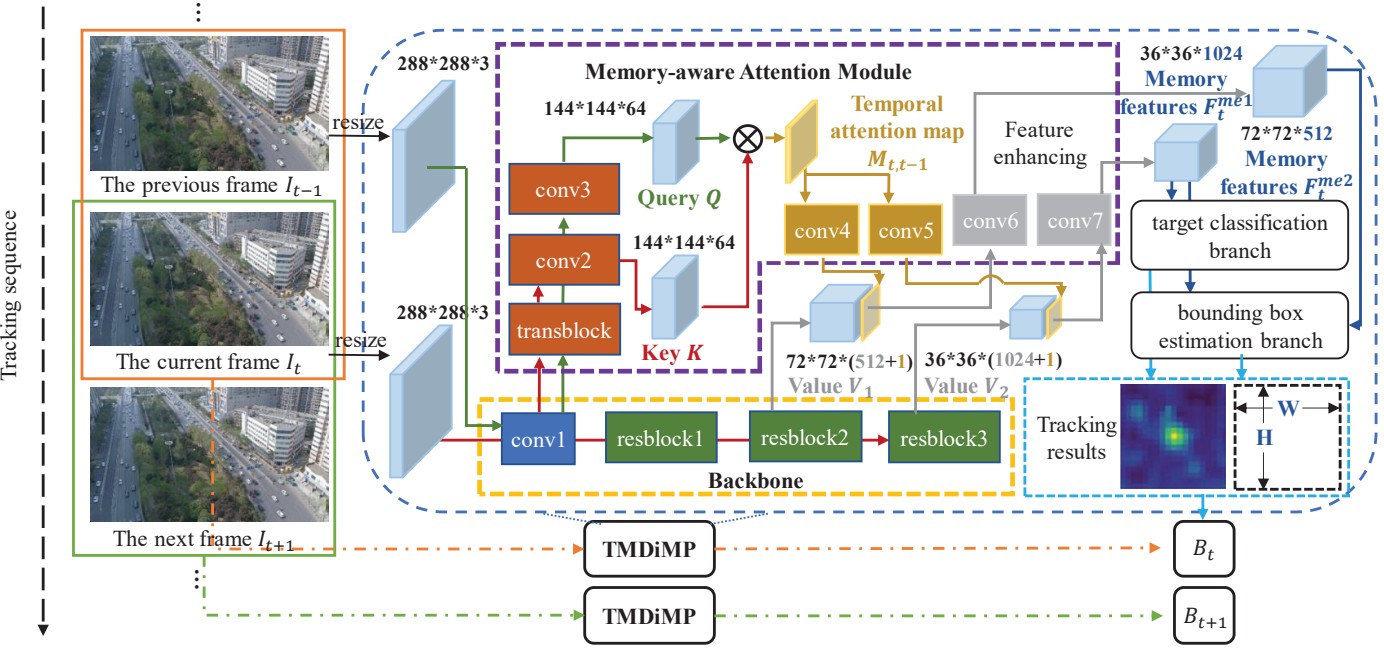

**Figure 1.** The framework of our proposed TMDiMP.

---

**Algorithm 1:** The online tracking process of our TMDiMP.

**Input:** A video sequence containing $nframe$ frames and the location of the target in the first frame.

**Output:** The locations of the target in each frame of the video sequence.

1 **for** $t = 2$ *to* $nframe$ **do**

2      step 1. Employ the backbone to extract low-level features $F_t^{LL}$ of the current frame $I_t$;

3      step 2. Employ the backbone to extract low-level features $F_{t-1}^{LL}$ of the previous frame $I_{t-1}$;

4      step 3. Employ the memory-aware attention module to calculate the temporal attention map $M_{t,t-1}$ according to $F_t^{LL}$ and $F_{t-1}^{LL}$;

5      step 4. Employ the backbone to extract high-level features $F_t^{HL}$ of the current frame $I_t$;

6      step 5. Employ the memory-aware attention module to generate memory features $F_t^{me}$ of the current frame $I_t$ using the temporal attention map $M_{t,t-1}$;

7      step 7. Employ the target classification branch to predict the location center of the target according to memory features $F_t^{me}$;

8      step 8. Employ the bounding box esitimation branch to predict the size of the target according to memory features $F_t^{me}$;

9      step 9. Determine the location of the target in the $t^{th}$ frame by the predicted location center and size;

10 **end**

---

In detail, the proposed TMDiMP takes a pair of adjacent frames as the backbone input. Here, we adopt ResNet50 pretrained on ImageNet as the backbone because ResNet50 can balance accuracy and speed during the tracking process.

First, temporal information crossing adjacent frames is calculated to obtain the temporal attention map. Specifically, the current frame $I_t$ and the previous frame $I_{t-1}$ are first processed by the *conv*1 in ResNet50 to obtain the low-level features $F_t^{LL}$ and $F_{t-1}^{LL}$. Then, $F_t^{LL}$ and $F_{t-1}^{LL}$ are fed into the carefully designed memory-aware attention module. A *transblock* and two convolutional layers, *conv*2 and *conv*3, are implemented to generate key $K$ and query $Q$, respectively. To avoid loss of location and boundary information, the transform operation does not change the size of low-level features. Then, the temporal attention map $M_{t,t-1}$ is calculated according to $K$ and $Q$.

Secondly, the memory features are generated by enhancing the high-level features of $I_t$ with $M_{t,t-1}$. As described above, the $M_{t,t-1}$ are first merged with high-level features to obtain semantic information. In TMDiMP, we extract features $F_t^{HL}$ of the current frame $I_t$ by *transblock*2 and *transblock*3 in ResNet50 as the values $V_1$ and $V_2$, respectively. Then, *conv*4 and *conv*5 are utilized for downsampling $M_{t,t-1}$ to the same size as $V_1$ and $V_2$. The downsampled attention maps are denoted by $M_{t,t-1}^{DS1}$ and $M_{t,t-1}^{DS2}$. We generate two values $V_1$ and $V_2$ because the employed bounding box estimation branch can cope with multiscale features to obtain a more accurate target size. Then, we merge $M_{t,t-1}^{DS1}$ and $M_{t,t-1}^{DS2}$ with $V_1$ and $V_2$ by an adaptive method to obtain memory features. We have

$$\begin{cases} F_t^{me1} = conv6(concat(M_{t,t-1}^{DS1}, V_1)) \\ F_t^{me2} = conv7(concat(M_{t,t-1}^{DS2}, V_2)), \end{cases} \tag{5}$$

where *concat* denotes the concatenate operation.

Finally, the tracking results are predicted by the two prediction branches according to memory features. The bounding box estimation branch takes $F_t^{me1}$ and $F_t^{me2}$ to estimate the width and height of the target. The target classification branch takes $F_t^{me2}$ to predict the score map of the current frame, which contains the location center of the target. The

tracking results can be determined by the size and location center of the target. For more details of the two branches, refer to DiMP [7].

The detailed architecture of our TMDiMP is shown in Table 2. The memory features are more discriminative and robust than the original image features so that the predicted branches can utilize the memory features to estimate more accurate results compared with the baseline, which is demonstrated by the conducted experiments.

**Table 2.** The detailed architecture of our TMDiMP.

| Backbone(Resnet50) | | | |
|---|---|---|---|
| **Name** | **Setting** | **Input** | **Output** |
| input | - | $\begin{bmatrix} 288 \times 288 \times 3 \\ 288 \times 288 \times 3 \end{bmatrix}$ | - |
| conv1 | $7 \times 7 \times 64$ | $288 \times 288 \times 3$ | $144 \times 144 \times 64$ |
| resblock1 | $\begin{bmatrix} 1 \times 1 \times 64 \\ 3 \times 3 \times 64 \\ 1 \times 1 \times 256 \end{bmatrix} \times 3$ | $144 \times 144 \times 64$ | $144 \times 144 \times 256$ |
| resblock2 | $\begin{bmatrix} 1 \times 1 \times 128 \\ 3 \times 3 \times 128 \\ 1 \times 1 \times 512 \end{bmatrix} \times 4$ | $144 \times 144 \times 256$ | $72 \times 72 \times 512$ |
| resblock3 | $\begin{bmatrix} 1 \times 1 \times 256 \\ 3 \times 3 \times 256 \\ 1 \times 1 \times 1024 \end{bmatrix} \times 6$ | $72 \times 72 \times 512$ | $36 \times 36 \times 1024$ |
| Memory-aware attention module | | | |
| Name | Setting | Input | Output |
| transblock | $\begin{bmatrix} 3 \times 3 \times 64 \\ 3 \times 3 \times 64 \end{bmatrix}$ | $144 \times 144 \times 64$ | $144 \times 144 \times 64$ |
| conv2 | $1 \times 1 \times 64$ | $144 \times 144 \times 64$ | $144 \times 144 \times 64$ |
| conv3 | $1 \times 1 \times 64$ | $144 \times 144 \times 64$ | $144 \times 144 \times 64$ |
| multiplication | $conv2 \otimes conv3$ | $\begin{bmatrix} 144 \times 144 \times 64 \\ 144 \times 144 \times 64 \end{bmatrix}$ | $144 \times 144 \times 64$ |
| softmax | - | $144 \times 144 \times 64$ | $144 \times 144 \times 64$ |
| reshape | - | $144 \times 144 \times 64$ | $144 \times 144 \times 1$ |
| conv4 | $3 \times 3 \times 1$ | $144 \times 144 \times 1$ | $72 \times 72 \times 1$ |
| conv5 | $3 \times 3 \times 1$ | $72 \times 72 \times 1$ | $36 \times 36 \times 1$ |
| concatenate1 | - | $\begin{bmatrix} 72 \times 72 \times 512 \\ 72 \times 72 \times 1 \end{bmatrix}$ | $72 \times 72 \times 513$ |
| concatenate2 | - | $\begin{bmatrix} 36 \times 36 \times 1024 \\ 36 \times 36 \times 1 \end{bmatrix}$ | $36 \times 36 \times 1025$ |
| conv6 | $1 \times 1 \times 512$ | $72 \times 72 \times 513$ | $72 \times 72 \times 512$ |
| conv7 | $1 \times 1 \times 1024$ | $36 \times 36 \times 1025$ | $36 \times 36 \times 1024$ |

## 4. The VIPUOTB Dataset

In this section, we describe our data collection and annotation, present various statistics compared with other public datasets and showcase different aspects of our dataset.

### 4.1. Data Collection and Annotation

A challenging UAV tracking dataset called VIPUOTB, captured by UAV cameras mounted on a DJI drone, is collected in urban scenes, where the road conditions are complex and the number of targets is large, especially small targets and similar targets. These targets are also denser and closer to each other in urban scenes. More than three domain experts who are students working in this field for more than one year annotated over 16,000 individual frames of 50 video sequences by using LableImg software. Figure 2 shows some frames of video sequences with annotated small objects. Due to the small size of the objects, we used object areas instead of whole images in subjective comparison. An example of obtaining an object area is illustrated in Figure 3.

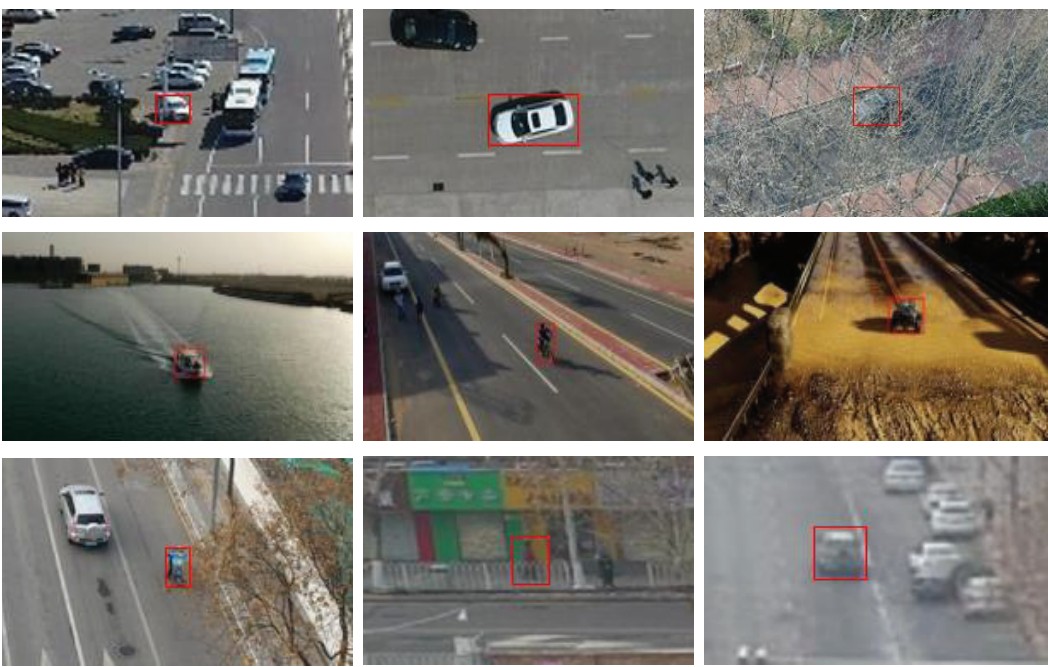

**Figure 2.** Due to the small size of objects, we use object areas instead of whole images to show differences of objects in UAVDT, UAV123, and VIPUOTB. We find that there are only various vehicles captured from real scenes in sequences from the UAVDT dataset shown in the first row. The sequences in UAV123 contain more comprehensive object categories and attributes. However, we can observe from the second row that there is a flat background and slight changes in most sequences of UAV123. Samples from our VIPUOTB are shown in the third row. We can see that the VIPUOTB dataset contains different kinds of objects, such as bicycles, pedestrians, and vehicles, compared with UAVDT. The backgrounds of sequences in our dataset are more complex than those of UAV123.

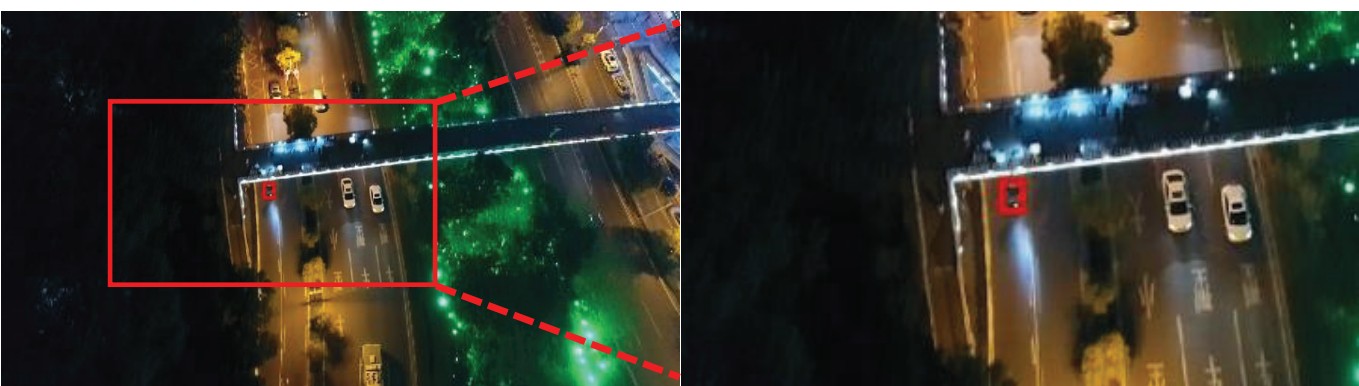

**Figure 3.** An example of obtaining an object area.

We find that the ground truth will be affected by the subjective factors of different experts. Taking Figure 4 as an example, the red bounding box and green bounding box are given by two experts. However, the red box labels the whole person whereas the green box labels the main part of the body. To ensure consistency, we ask domain experts $De = 3$ to annotate each video clip and the same expert to annotate consecutive frames to avoid subjective factors on small objects. The final annotated ground truth is obtained by the agreement $B$,

$$B = (C, W, H), \qquad (6)$$

where $C = \frac{1}{De} \sum_{i=1}^{De} c_i$, $W = \frac{1}{De} \sum_{i=1}^{De} w_i$, and $H = \frac{1}{De} \sum_{i=1}^{De} h_i$. $c$, $w$, and $h$ are the location center, width, and height of the object, respectively.

To ensure annotation quality, we randomly checked the annotation results on 300 samples extracted from different video clips of two Ph.D. students whose research area is object tracking and quickly revised them. We repeat the above error correction work three times.

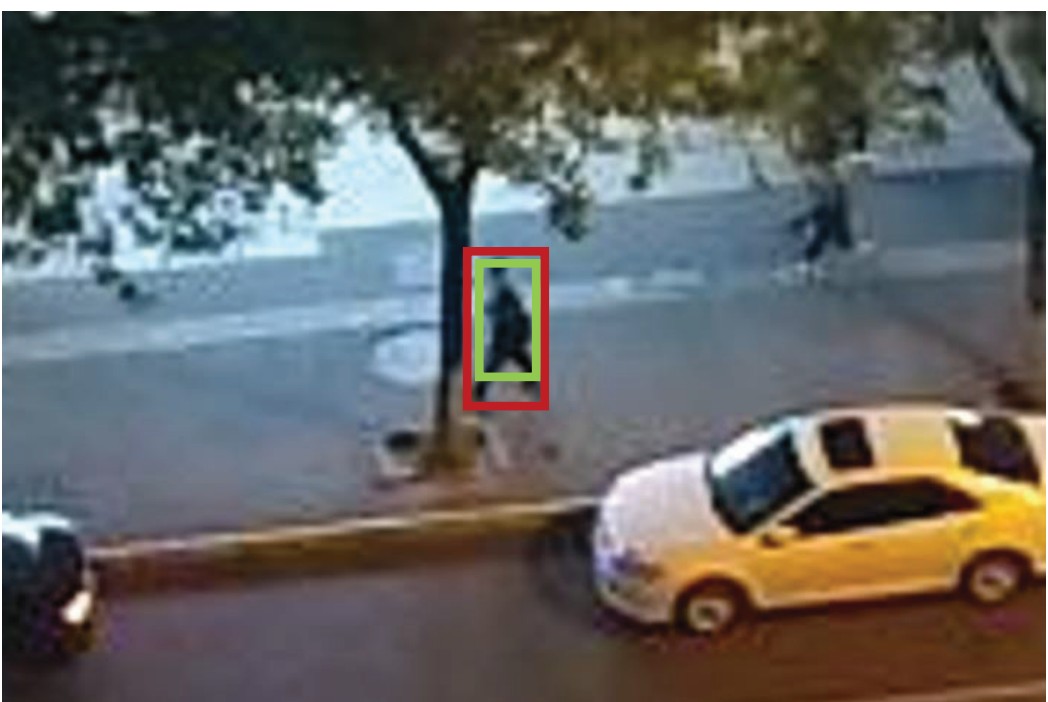

**Figure 4.** Subjective factors in the labeling process.

Statistics on VIPUOTB and two famous UAV datasets are summarized in Table 3. Four important objective criteria, types of attributes, categories of objects, average proportion of the target size to an image (APTS) and average moving distance between adjacent frames during camera motion (AMDAF), are used to measure the diversity of the datasets. The details about APTS and AMDAF are given in Sections 4.2 and 4.3.

**Table 3.** VIPUOTB statistics and the comparison results with other UAV datasets.

|  | UAVDT | UAV123 | VIPUOTB |
|---|---|---|---|
| Vision task | SOT/MOT | SOT | SOT/MOT |
| Number of videos (SOT/MOT) | 50/50 | 123 | 50/50 |
| Types of attributes (SOT) | 8 | 12 | 14 |
| Categories of objects (SOT) | 1 | 8 | 5 |
| APTS (SOT) | 0.602% | 0.764% | 0.055% |
| AMDAF (SOT) | 4.46 | 5.82 | 15.88 |

*4.2. Normalized Object Size*

We compare the APTS among these three datasets because the small size of objects is one of the important factors we focus on. APTS can be calculated according to Equation (7),

$$APTS = \frac{1}{N} \sum_{n=1}^{N} \frac{w_{object}^n \times h_{object}^n}{w_{image}^n \times h_{image}^n},$$

(7)

where $N$ is the number of total frames in a video sequence, $w_{object}^n$ and $h_{object}^n$ denote the width and height of an object in the $n$th frame, and $w_{image}^n$ and $h_{image}^n$ denote the width

and height of the *n*th frame. We find that the average sizes of the objects in UAVDT and UAV123 are approximately the same, which are more than ten times larger than those in our database.

### 4.3. Fast Camera Motion

To compare the camera motion speed on different datasets, we manually select all frames when the camera motion occurred. The camera motion speed can be reflected by the average moving distance between adjacent frames. The AMDAF can be calculated according to Equation (8),

$$AMDAF = \frac{1}{KM} \sum_{k=1}^{K} \sum_{m=1}^{M_k} \sqrt{\left(c_x^m - c_x^{m-1}\right)^2 + \left(c_y^m - c_y^{m-1}\right)^2}, \tag{8}$$

where *K* is the total number of video clips containing camera motion. $M_k$ is the frame numbers of the *k*th clip. $c_x^m$, and $c_y^m$ are the abscissa and ordinate of the target center in the *m*th frame. The AMDAF of VIPUOTB is the largest among the three datasets, which are approximately three times those of UAVDT and UAV123.

### 4.4. Attributes

A summary of 14 tracking attributes presented in our proposed VIPUOTB dataset is shown in Table 4. We define two new attributes according to APTS and AMDAF, which are NSO and FCM. In addition, during the observation from the real road environment, we find that there are two specific attributes, FS and MS, which will seriously affect the tracking performance. The distribution of these attributes over our dataset is shown in Figure 5 and Figure 5a shows the number of sequences with different attributes. Figure 5b shows the number of sequences with different numbers of attributes. From Figure 5a, we can observe that some attributes occur more frequently, such as SO and CM. In particular, more than 60% of videos contain small-object and camera motion cases, which we mainly focus on in this paper, and among them, 79.1% of the sequences have no less than four challenge factors. We can summarize from Figure 5b that 50% of the sequences have no less than five challenge factors over the whole dataset.

**Table 4.** Fourteen different attribute definitions in VIPUOTB.

| Attribute | Abbreviation | Definition |
| --- | --- | --- |
| Small Object | SO | The target box is smaller than $30 \times 30$ pixels in at least one frame. |
| Normalized Small Object | NSO | The APTS of a sequence is smaller than 0.1%. |
| Camera Motion | CM | Abrupt motion of the camera. |
| Fast Camera Motion | FCM | The AMDAF of a sequence is larger than 15 pixels. |
| Scale Variation | SV | The ratio of bounding box is outside the range [0.5, 2]. |
| Illumination Variation | IV | The illumination in the target region changes. |
| Object Blur | OB | The target region is blurred due to target or camera motion. |
| Background Clutter | BC | The background near the target has similar appearance as the target. |
| Large Occlusion | LO | The target is partially or fully occluded in the sequence. |
| Aspect Ratio Changing | ARC | The ratio of bounding box aspect ratio is outside the range [0.5, 2]. |
| Fast Motion | FM | Motion of the target is larger than 20 pixels between adjacent frames. |
| Night | NI | The data are collected at night. |
| Fewer Similar object | FS | There are no more than five similar targets around the target. |
| Multi Similar object | MS | There are more than five similar targets around the target. |

Seven attributes, SO, CM, BC, ARC, IV, SV, and LO are common to the three datasets VIUOTB, UAVDT, and UAV123. Except for these common attributes, our VIPUOTB contains NSO, FCM, FM, NI, FS, and MS, which are not exciting in UAVDT. Compared with UAV123, the difference is that the attributes of out-of-view (OV) and viewpoint change (VC) are not considered in our VIPUOTB, both of which have recently been regarded as hot research fields. In the future, we will add more video clips with OV and CV attributes to our

datasets and improve the performance of our proposed method on these two attributes while maintaining the performance of other attributes.

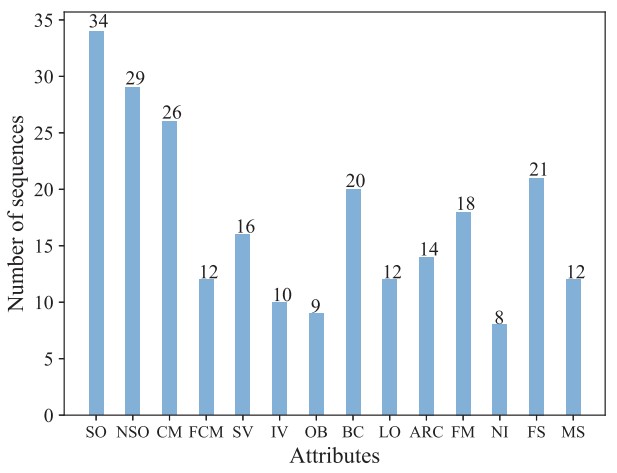

(**a**) Number of sequences with different attributes

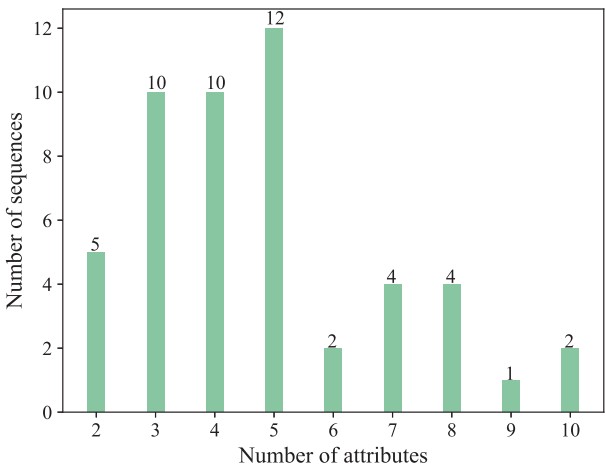

(**b**) Number of sequences with different number of attributes

**Figure 5.** Attribute distribution of VIPUOTB.

### 4.5. Categories

There are five object categories—pedestrian, car, bus, UAV, and bicycle—in our VIPUOTB dataset, all of which frequently appear in the city scene. The UAVDT dataset consists of cars, trunks, and buses. The bird, building, and wakeboard in UAV123 are not considered in our VIPUOTB, which are not as important and common as other categories for object tracking in city scenes.

### 4.6. Object Location

Figure 6 presents the heatmaps of object location by superimposing all the binary maps consisting of the object ground truth and the background. The lighter the color is in a heatmap, the more frequently the object appears. Figure 6a–c are location maps of all the objects on UAV123, UAVDT, and VIPUOTB, respectively. We can observe from Figure 6 that the objects in UAVDT and UAV123 are mainly located in the center of an image. This centralized distribution will cause the problem that deep neural networks learn strong center bias as a priori knowledge in the training stage [13,52]. In contrast to the location distribution of UAVDT and UAV123, the objects in VIPUOTB are evenly distributed in different locations. These phenomena show that our dataset has higher diversity in location than the two other datasets.

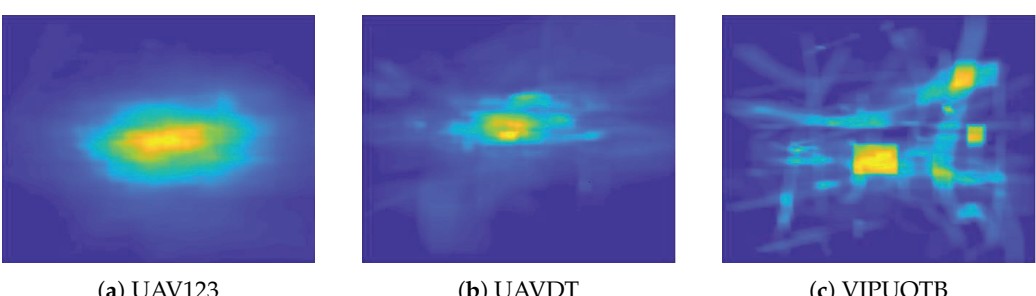

(**a**) UAV123        (**b**) UAVDT        (**c**) VIPUOTB

**Figure 6.** The location heatmaps of different datasets.

### 5. Experiments

In this section, we perform an extensive evaluation of our proposed TMDiMP through several experiments. First, we discuss the important parameter settings of the network.

Secondly, attribute-based evaluation is given to verify the effectiveness of TMDiMP in different situations. Thirdly, the experimental performance of our proposed method is described on four UAV tracking benchmarks by comparing it to state-of-the-art methods, including MDNet [9], STRCF [39], ECO-HC and ECO [40], VITAL [41], ATOM [42], KYS [28], DiMP [7], PrDiMP [8], AUTO [22], TrDiMP [29], and STM [30]. Fourth, we prove the effectiveness of the memory-aware attention module utilized in our method. All the methods were implemented by PyTorch installed on a PC with i7-11700k and RTX 3090.

The tracking performance was measured by the precision, normalized precision and success of one-pass evaluation (OPE) [53]. The precision is computed by using the center location error (CLE) between the estimated location and the ground truth. Because the precision metric is sensitive to target size and image resolution, the normalized precision metric is proposed in [32] to address this problem. Different trackers are ranked with precision and normalized precision metrics by using a threshold (20 pixels) and an area under the curve (AUC) between 0 and 0.5, respectively. The success is computed by using the intersection over union (IoU) of the estimated bounding box and ground truth. The tracking algorithms are ranked with a success metric by using an AUC between 0 and 1. The complete code and our dataset will be released upon publication.

### 5.1. Parameter Settings

#### 5.1.1. The Level of Features

As we know, the low-level features contain a greater amount of location and boundary information, and feature extraction in the low-level stage requires fewer computational resources. Therefore, we use features obtained from the *conv*1 layer to calculate the attention map of $Q$ and $K$. We conduct an experiment to prove which features extracted from *conv*1 and *resblock*1 of ResNet50 are more representative. The comparison results are recorded in Table 5. The tracking performance estimated by using attention maps calculated according to features from *conv*1 is better than *resblock*1 on four metrics, including precision, normalized precision, success, and FPS. In addition, the tested FPS values are different for two reasons. First, the data preprocessing time is different because of the different sizes of images in the four datasets. Secondly, if targets are missed by trackers, discriminative online learning trackers will redetect targets, which increases the time consumption of online tracking.

**Table 5.** The comparison tracking results of utilizing different attention maps.

| Dataset | VIPUOTB | | UAVDT | | UAV123 | | VisDrone | |
|---|---|---|---|---|---|---|---|---|
| Level of Features | Conv1 | Resblock1 | Conv1 | Resblock1 | Conv1 | Resblock1 | Conv1 | Resblock1 |
| Precision | 91.9% | 87.7% | 83.3% | 79.7% | 85.7% | 83.4% | 85.0% | 79.5% |
| Normalized precision | 86.2% | 82.1% | 72.8% | 66.6% | 80.8% | 76.9% | 79.5% | 74.3% |
| Success | 70.1% | 65.9% | 62.9% | 58.0% | 65.0% | 62.3% | 64.3% | 60.5% |
| FPS | 35.5 | 34.0 | 47.2 | 47.2 | 45.6 | 45.1 | 40.2 | 38.9 |

#### 5.1.2. The Number of Frames Applied in Memory-Aware Attention Module

The number of frames taken by the memory-aware attention module needs to be determined manually in our method. A single frame can only provide insufficient information. In contrast, more valueless information will introduce additional noise. We use the memory features accumulated by different numbers of frames to estimate tracking results, which are recorded in Table 6. Note that different numbers of frames do not affect the time consumption of the algorithm because only low-level features are extracted to calculate the attention map. We can see that the values of precision, normalized precision and success decrease as the number of frames used increases. Two consecutive frames can retain the most effective temporal memory. Therefore, we choose adjacent frames as the input of our memory-aware attention module.

**Table 6.** The comparison tracking results of utilizing different numbers of frames.

| Dataset | VIPUOTB | | UAVDT | | UAV123 | | VisDrone | |
|---|---|---|---|---|---|---|---|---|
| Number of Frames | 2 | 3 | 2 | 3 | 2 | 3 | 2 | 3 |
| Precision | 91.9% | 86.4% | 83.3% | 79.3% | 85.7% | 83.5% | 85.0% | 84.2% |
| Normalized precision | 86.2% | 81.8% | 72.8% | 66.9% | 80.8% | 77.2% | 79.5% | 78.6% |
| Success | 70.1% | 66.6% | 62.9% | 58.1% | 65.0% | 62.6% | 64.3% | 63.3% |

### 5.2. Attribute-Based Evaluation

To further explore the effectiveness of the TMDiMP tracker on different situations, we also estimate them on all attributes in Figure 7. We find that almost all trackers cannot achieve the same performance on NSO and FCM as SO and CM, which indicates that the NSO and FCM are more challenging than SO and CM. As shown in Figure 7a–d, our TMDiMP shows the best performance on attributes of SO and NSO, and the second-best performance on CM and FCM, which we mainly focus on in this paper. The success values of DiMP are 3.9%, 1.3%, 2.7%, and 0.5% lower than those of TMDiMP on these four challenging attributes.

In general, our method can obtain competitive results on most attributes. However, the performance is unsatisfactory when TMDiMP addresses the attributes of night and object blur. Through analysis, we find that video clips with object blur are captured mostly at night. Unfortunately, our method cannot extract discriminative features in a dark environment. We will pay more attention to these two attributes in the future.

### 5.3. State-of-the-Art Comparisons

The performance of our proposed model is evaluated on both objective and subjective evaluations.

#### 5.3.1. VIPUOTB

The overall performance for all tracking methods on VIPUOTB is reported by the precision, normalized precision, and success plots of OPE, as shown in Figure 8. TMDiMP outperforms all state-of-the-art methods on all precision, normalized precision, and success metrics. Our method improves over the baseline DiMP on the precision, normalized precision, and success of OPE by 4.8%, 2.3% and 2.9%, respectively.

#### 5.3.2. UAVDT

Figure 9 illustrates the precision, normalized precision, and success plots among all competitors on UAVDT, and the performance score for each tracker is given in the legend of the figure. The proposed TMDiMP method performs favorably, with a precision value of 83.3%, normalized precision value of 72.8%, and success value of 62.9%. Compared with the baseline DiMP, our method improves the precision and success of OPE by 4.6%, 5.5%, and 5.0%, respectively. Although TrDiMP has the best precision performance of 84.9%, STM has the best normalized precision performance of 76.7% and the best success performance of 64.7% among all the methods, which is 1.6%, 3.9% and 1.8% higher than those of our proposed model. Through the observation and analysis of the results, we find that our tracker usually fails to track objects that are occluded for a long time. Similar objects are usually selected by TMDiMP under this situation, which is illustrated in the following Section 6.

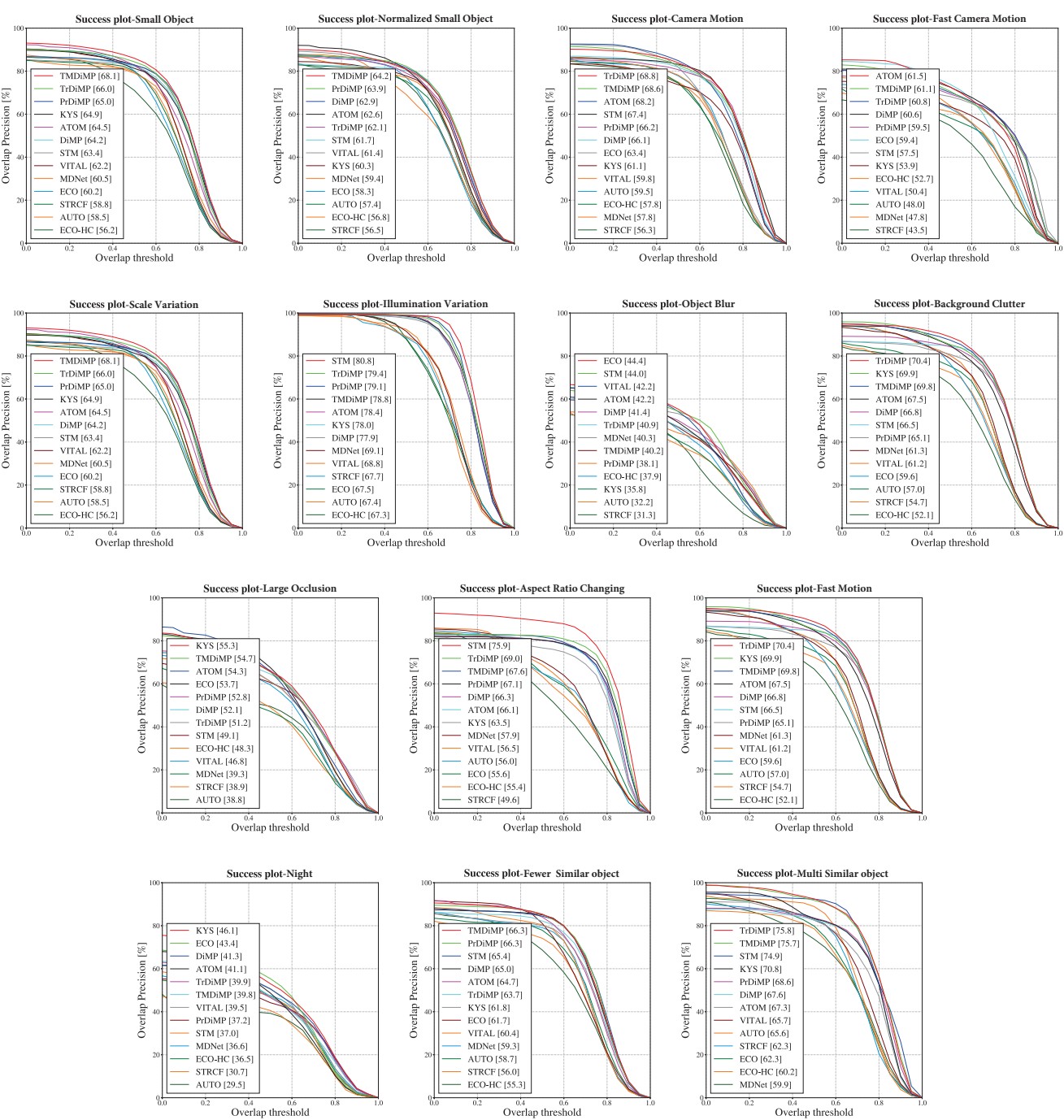

**Figure 7.** Success plots of different attributes on VIPUOTB.

### 5.3.3. UAV123

We also compare all competitors on UAV123 and illustrate the precision, normalized precision, and success plots in Figure 10. The highest precision value of 87.6%, normalized precision value of 83.0%, and success value of 66.8% are achieved by PrDiMP. Our model ranks second in all the precision, normalized precision and success metrics, which have approximately the same performance as baseline DiMP. Similar to UAVDT, the failures are caused by long-term occluded targets. We also give the false example in Section 6.

### 5.3.4. VisDrone

The performance for each tracker on the VisDrone dataset is exhibited in Figure 11. Our tracker TMDiMP achieves the best performance, with a precision score of 85.0%, normalized precision score of 79.5%, and success score of 64.3%. It surpasses ECO and STM by 1.8%, 0.7%, and 1.0% in the precision, normalized precision, and success plots, respectively. In addition, our tracker has a relative gain of 4.2% precision, 3.7% normalized precision, and 3.0% success over DiMP.

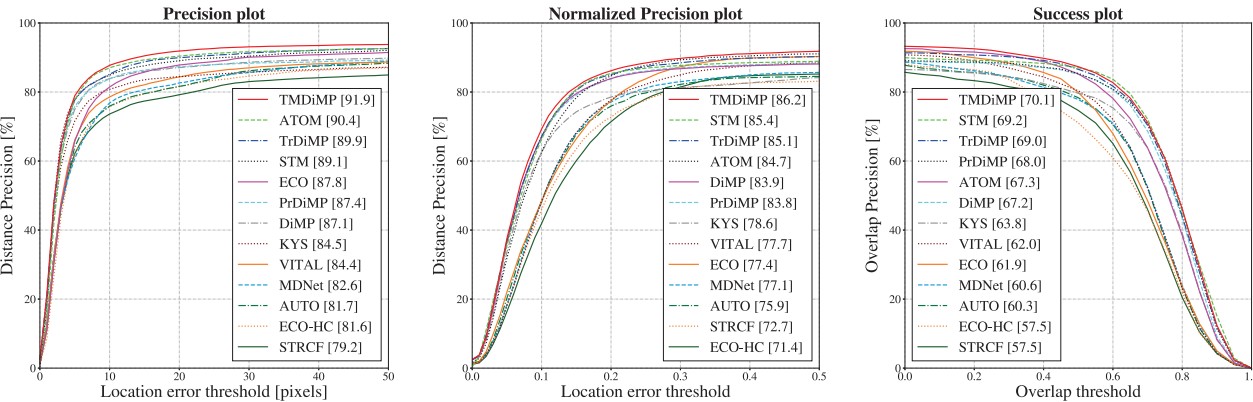

**Figure 8.** Precision, normalized precision and success plots of OPE on VIPUOTB.

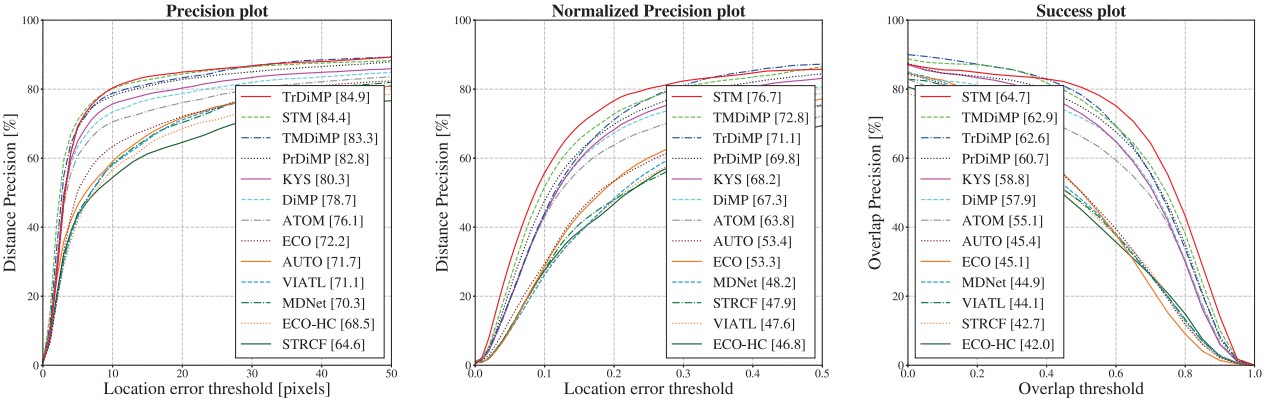

**Figure 9.** Precision, normalized precision, and success plots of OPE on UAVDT.

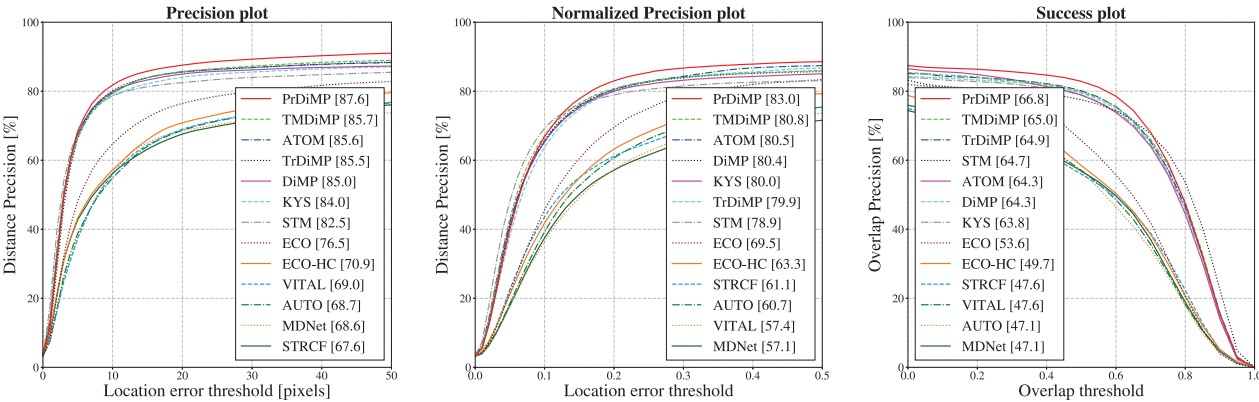

**Figure 10.** Precision, normalized precision, and success plots of OPE on UAV123.

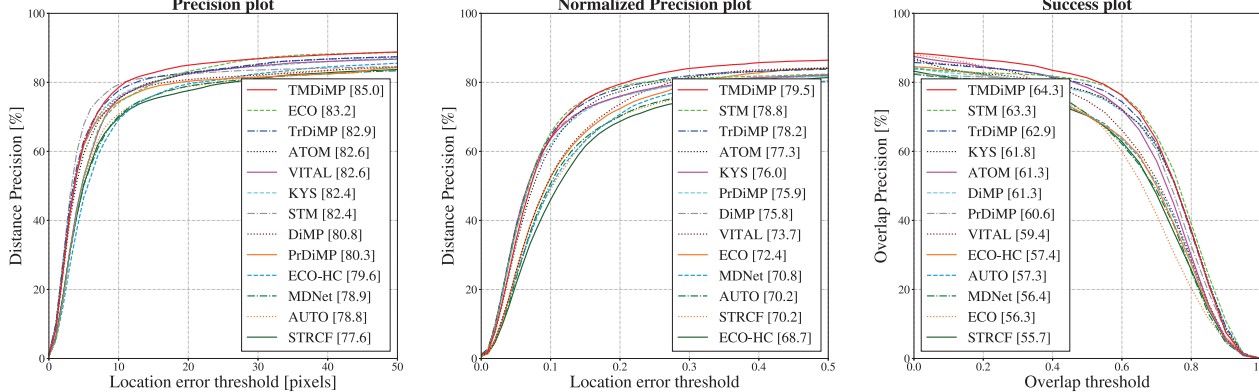

**Figure 11.** Precision, normalized precision, and success plots of OPE on VisDrone.

### 5.3.5. Average CLE and IoU

Table 7 reports the average CLE and IoU of all compared trackers on four benchmarks. This shows that only TMDiMP can achieve at least the top three results on all datasets.

### 5.3.6. Subjective Comparison

To provide a more intuitive exhibition, the subject assessment results obtained from the top seven trackers (TMDiMP, TrDiMP, STM, PrDiMP, DiMP, ATOM, and ECO) on a challenging sequence in our VIPOUTB database are illustrated in Figure 12.

**Table 7.** Overall evaluation of the four datasets. The top three trackers are marked by red, green, and blue colors. Avg. CLE and Avg. IoU denote average CLE and average IoU, respectively.

|  | **VIPUOTB** | | **UAVDT** | | **UAV123** | | **VisDrone** | |
|---|---|---|---|---|---|---|---|---|
|  | **Avg. CLE** | **Avg. IoU** | **Avg. CLE** | **Avg. IoU** | **Avg. CLE** | **Avg. IoU** | **Avg. CLE** | **Avg. IoU** |
| STRCF | 44.65 | 0.58 | 74.24 | 0.43 | 95.89 | 0.49 | 53.32 | 0.56 |
| MDNet | 32.54 | 0.61 | 59.16 | 0.45 | 105.74 | 0.48 | 64.30 | 0.57 |
| VITAL | 40.29 | 0.63 | 61.69 | 0.44 | 91.52 | 0.49 | 45.52 | 0.60 |
| ECO | **18.66** | 0.63 | 52.76 | 0.45 | 81.60 | 0.55 | 52.13 | 0.57 |
| ECO-HC | 44.68 | 0.58 | 74.61 | 0.42 | 91.41 | 0.51 | 57.82 | 0.58 |
| ATOM | 28.83 | 0.68 | 51.27 | 0.56 | **46.89** | 0.66 | 33.90 | 0.62 |
| AUTO | 39.70 | 0.61 | 68.39 | 0.46 | 99.52 | 0.48 | 69.34 | 0.58 |
| DiMP | 28.76 | 0.68 | 44.49 | 0.59 | 53.58 | 0.66 | 34.09 | 0.62 |
| PrDiMP | 33.74 | 0.69 | 37.46 | 0.62 | **34.25** | **0.69** | **27.79** | 0.62 |
| KYS | 91.36 | 0.65 | 45.97 | 0.60 | 59.78 | 0.66 | 32.69 | 0.63 |
| TrDiMP | 23.89 | **0.70** | **34.63** | **0.64** | 50.63 | **0.67** | **30.00** | **0.64** |
| STM | **21.75** | **0.70** | **42.03** | **0.66** | 55.14 | **0.67** | 55.25 | **0.64** |
| TMDiMP | **20.55** | **0.71** | **35.14** | **0.64** | **47.47** | **0.67** | **21.31** | **0.65** |

This sequence exhibits three challenging factors: a tiny bicycle, camera motion, and severe occlusion by trees. Because the proportion of target pixels to the total pixels of an image is too small, trackers are easily disturbed by the cluttered background. We can observe from the 40th frame that the pedestrian begins to cycle through the trees. The PrDiMP misses this bicycle in the 56th frame after the target passes through trees. At the 193rd frame, the camera mounted on the UAV moves suddenly to capture the target, and all other trackers tend to drift away from the bicycle temporarily. Our TMDiMP finds the object again when the camera is stable, which can be found in the 203rd frame. This experiment illustrates that the proposed memory-aware attention mechanism can encourage our tracker to learn the pattern of camera motion.

We also compare the subjective results of the top seven trackers (TMDiMP, TrDiMP, STM, VITAL, KYS, ATOM, and ECO) on a sequence from the public dataset VisDrone,

which is shown in Figure 13. The tracked target is a moving pedestrian. At the beginning of the sequence, the pedestrian is walking through a flag. All trackers basically lose the target when the target is occluded. Only our TMDiMP retracks the pedestrian.

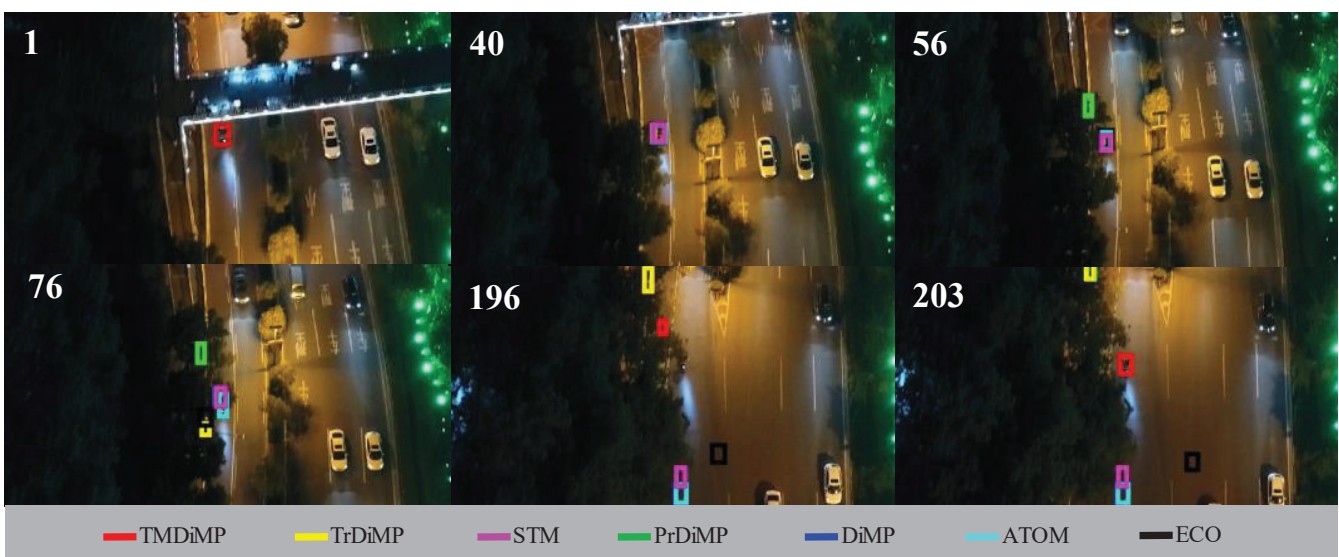

**Figure 12.** The subjective results in VIPUOTB.

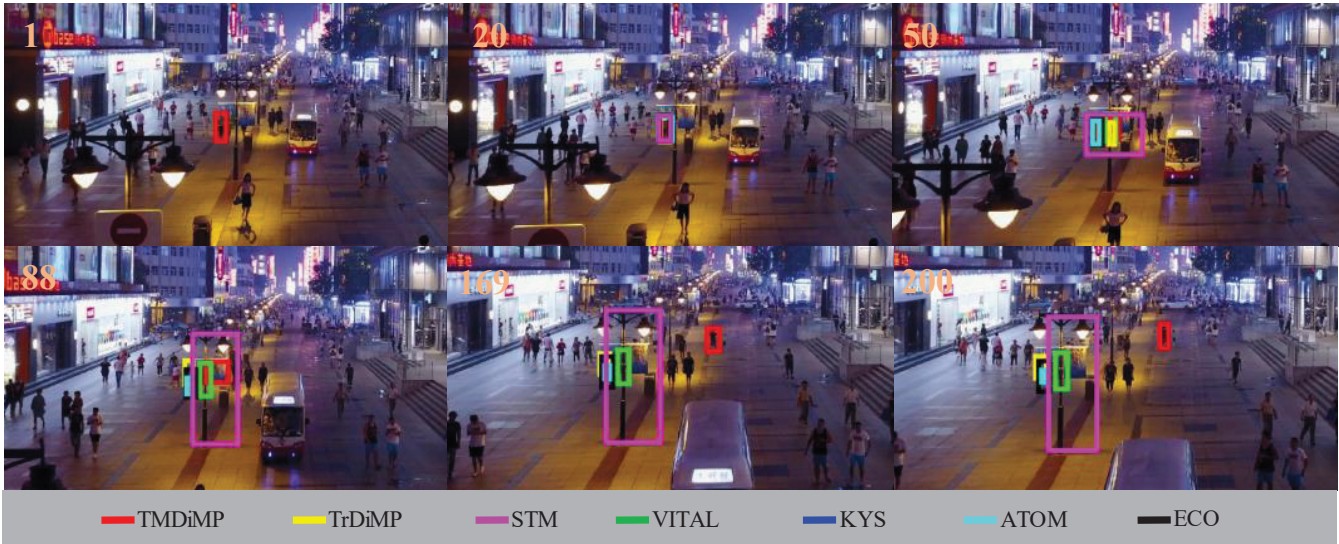

**Figure 13.** The subjective results in VisDrone.

### 5.3.7. Score Map Visualization

As introduced in Section 3, our TMDiMP and baseline DiMP distinguish the target from the background by the target classification scores, which can determine the center location of the target. Figure 14 visualizes the tracking results and their corresponding score maps generated by DiMP and our TMDiMP. The yellow and red rectangles denote ground-truth bounding boxes and tracking results, respectively. The first column of Figure 14 gives the tracking results from the 76th to the 78th frame in a sequence, from which we can see that the DiMP tracker loses the target in the 77th frame. The corresponding score map of the 77th frame in the second column in Figure 14 exhibits two highlighted areas, and the lighter area is marked with a purple rectangle. In contrast, the fourth column shows the uniform highlighted area of three consecutive frames and our model can track the target object without being affected by the similar object on the surroundings, which indicates that the temporal memory can enhance features of small objects.

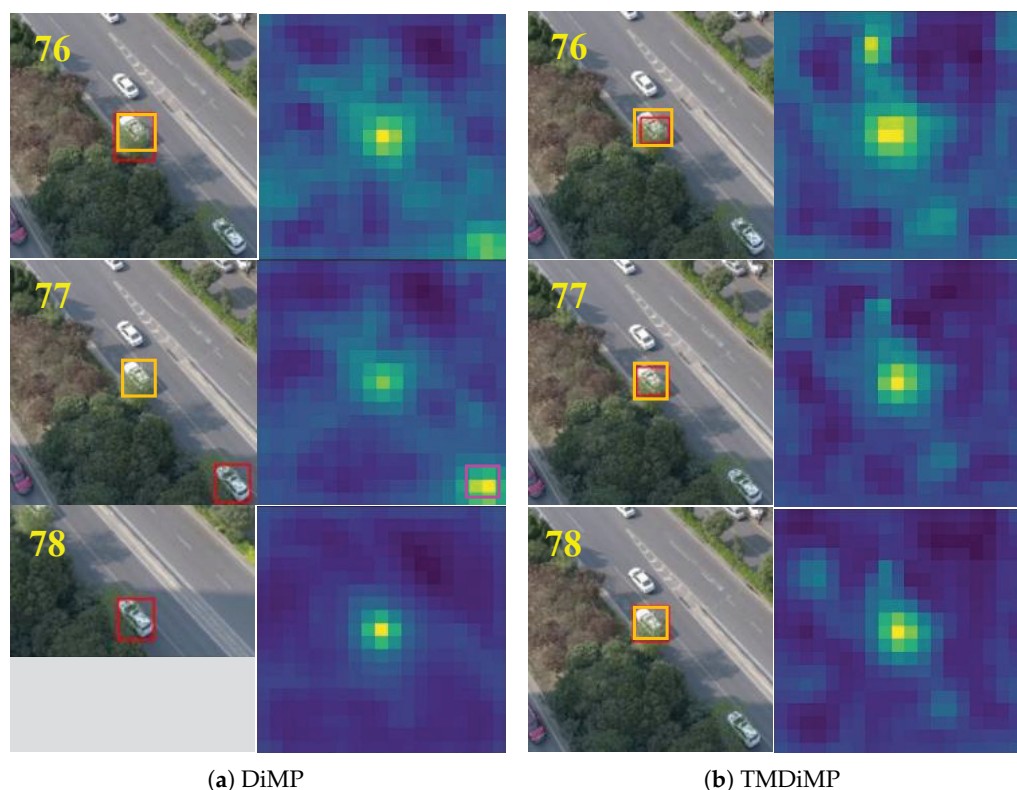

(**a**) DiMP            (**b**) TMDiMP

**Figure 14.** The score maps of DiMP and TMDiMP.

### 5.4. Ablation Study

In this paper, we present a novel memory-aware attention mechanism inspired by the classic self-attention [51]. In order to verify the effectiveness of our proposed method, we use the classic self-attention module instead of the memory-aware attention module in Figure 1 and test the modified framework on different datasets. In classic self-attention module, value $V$ is obtained by processing low-level features $F_t^{LL}$ of the current frame $I_t$ with a sample convolutional layer. Then, the enhanced low-level features $F_{SA}^{LL}$ are calculated by adding the product of value $V$ and temporal attention map $M_{t,t-1}$ to $V$. Finally, the concatenated residual blocks in backbone take $F_{SA}^{LL}$ to generate high-level features $F_{SA}^{HL}$. Different from the classic attention module, our proposed framework first extracts high-level features $F_t^{HL}$ of the current frame $I_t$ as values $V$. Then, the well-designed memory-aware attention module is employed to calculate memory features $F^{me}$ by enhancing values $V$ with the temporal attention map $M_{t,t-1}$. The tracking results predicted by using memory features $F^{me}$ and high-level features $F_{SA}^{HL}$ are recorded in Table 8, respectively, which suggests that our TMDiMP can utilize high-level features to generate more discriminative features mentioned in Section 3.

**Table 8.** The comparison results of utilizing different attention modules.

| Dataset | VIPUOTB | | UAVDT | | UAV123 | | VisDrone | |
|---|---|---|---|---|---|---|---|---|
| **Attention Modules** | **Proposed** | **Classic** | **Proposed** | **Classic** | **Proposed** | **Classic** | **Proposed** | **Classic** |
| Precision | 91.9% | 84.4% | 83.3% | 78.0% | 85.7% | 83.8% | 85.0% | 83.8% |
| Normalized precision | 86.2% | 80.5% | 72.8% | 66.8% | 80.8% | 77.8% | 79.5% | 77.7% |
| Success | 70.1% | 65.1% | 62.9% | 58.6% | 65.0% | 63.3% | 64.3% | 62.8% |

### 5.5. Implementation Details

The size of the adjacent frames is resized to 288 × 288. The baseline DiMP [7] and other previous works PrDiMP [8] and TrDiMP [29] utilize the training splits of LaSOT [31],

TrackingNet [32], GOT-10k [33], and COCO [54] for offline training. However, COCO only contains single images. We use ImageNetVid [55] instead of COCO in this work. Our framework is trained for 50 epochs with 2600 iterations per epoch and 10 image pairs per batch. The Adam optimizer [56] is employed with an initial learning rate of 0.01 and a decay factor of 0.2 every 15 epochs.

## 6. Discussion

In this section, we give failure cases of our TMDiMP in Figure 15, which shows three challenging video sequences in UAVDT (first row), UAV123 (second row), and VIPUOTB (third row). The ground-truth and tracking results of our method are marked in blue rectangles and red rectangles. In the first row of Figure 15, the target in UAVDT is occluded by a similar object at the beginning of the video, and TMDiMP tracks a similar object when two cars separate completely in the 224th frame. Similarly, our method tracks another person when the target person is occluded by a tent in UAV123. The third row of Figure 15 shows one of the most challenging video sequences in our dataset, which contains serious problems of object blur and occlusion. We can see from the eighth frame that the target car's appearance is blurred when it is passing through trees. Our TMDiMP misses the target in all the following frames of this video. In fact, all the state-of-the-art methods we tested fail to track the object in this situation.

We can conclude from these three failure cases that our tracker cannot handle long-term disappearances of targets in tracking. This problem may be addressed by improving the online training performance of the tracker. When targets disappear, the online training mechanism can guide trackers to continuously search for lost targets globally according to their history appearances. However, it is challenging to find representative history appearances because sometimes trackers give negative samples high confidence scores. In addition, the computational resources onboard UAV are limited, so the online training mechanism may affect the real-time performance of trackers. We will pay more attention to designing an efficient online training mechanism in our future work.

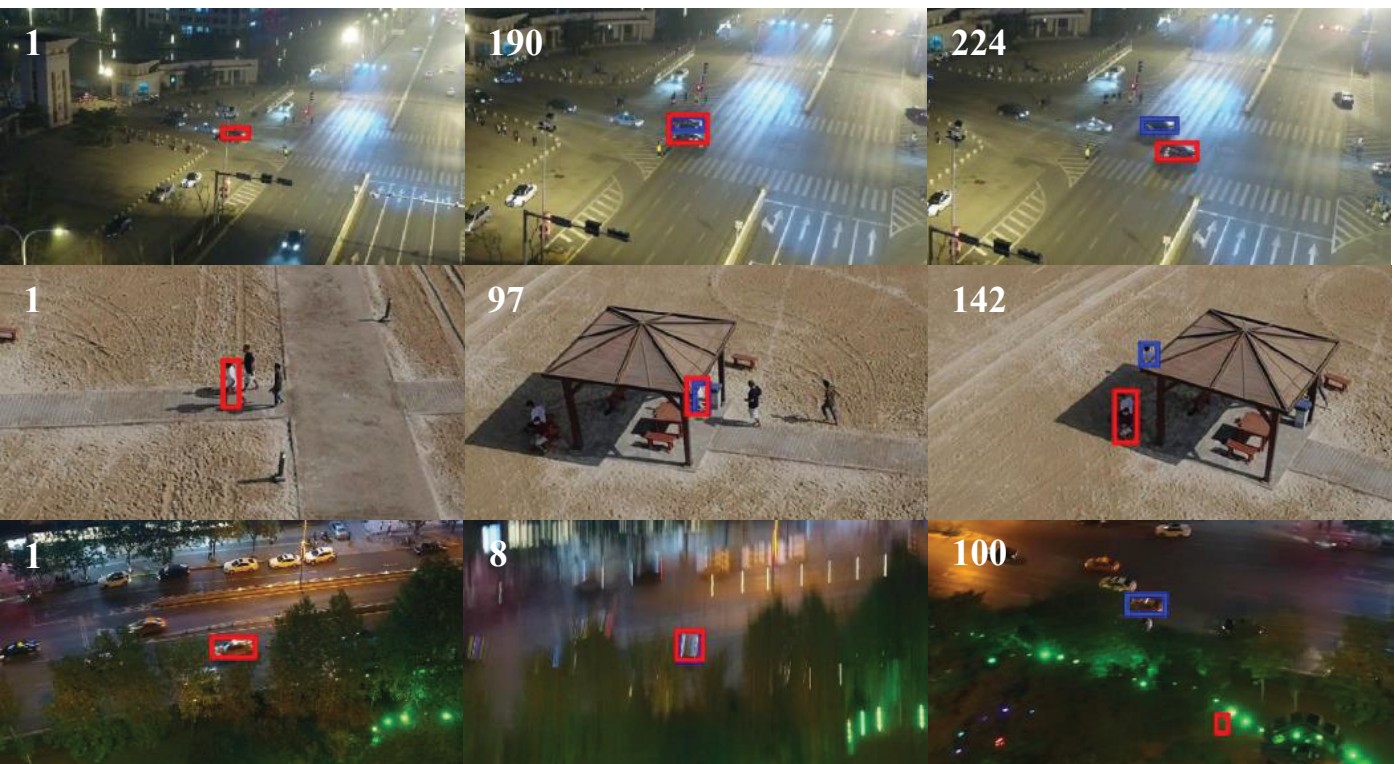

**Figure 15.** Failure cases in UAVDT and UAV123.

## 7. Conclusions

In this study, we focus on mitigating UAV object-tracking problems caused by small objects and camera motion by using advanced artificial intelligence technology. Thus, a novel tracker called TMDiMP is proposed. TMDiMP is a discriminative tracker with end-to-end training ability that utilizes a carefully designed memory-aware attention mechanism to generate more discriminative features of small objects and overcome the object-forgetting problem of camera motion. We also build a UAV object-tracking dataset, VIPUOTB, which is different from existing datasets in terms of object size, camera motion speed, location distribution, etc. Compared with other UAV object-tracking datasets, our VIPUOTB tracks the smallest objects and has the fastest camera motion speed. Various experiments, including parameter setting, attribute-based evaluation, objective comparison, subjective comparison, and ablation study are conducted to verify the effectiveness of our proposed method. Through the analysis of the experimental results, we conclude that our TMDiMP can achieve a better performance on our VIPUOTB dataset and three public datasets, UAV123, UAVDT, and VisDrone, compared to state-of-the-art methods.

The failure cases show that our tracker misses targets when they disappear for a long time. In the future, we will pay more attention to data with multiple challenging attributes, such as long-term object blur and occlusion. We will solve these problems by improving the online training performance of the tracker. In addition, we will expand our generated dataset constantly by adding more video sequences and attributes, such as out-of-view objects.

**Author Contributions:** Conceptualization, Z.Y. and B.H.; methodology, Z.Y.; software, Z.Y.; validation, Z.Y., B.H., W.C. and X.G.; formal analysis, Z.Y. and B.H.; investigation, Z.Y.; resources, B.H.; data curation, Z.Y.; writing—original draft preparation, Z.Y.; writing—review and editing, Z.Y. and B.H.; visualization, Z.Y.; supervision, B.H.; project administration, B.H.; funding acquisition, B.H. All authors have read and agreed to the published version of the manuscript.

**Funding:** This work was supported in part by National Natural Science Foundation of China, Grant No. 62076190, 41831072, 41874195 and 62036007, in part by The Key Industry Innovation Chain of Shaanxi, Grant No. 2022ZDLGY01-11.

**Data Availability Statement:** Four publicly available datasets LaSOT [31], TrackingNet [32], GOT-10k [33], and COCO [54] were used for the training of the proposed TMDiMP. Three publicly available datasets UAVDT [27], UAV123 [34], VisDrone [35], and a generated dataset, VIPUOTB, were used for the testing of the proposed TMDiMP.

**Conflicts of Interest:** The author declares no conflict of interest.

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
