# Peer review of "TMDiMP: Temporal Memory Guided Discriminative Tracker for UAV Object Tracking"

_remotesensing, doi:10.3390/rs14246351_

Round 1

Reviewer 1 Report

Present study "TMDiMP: Temporal memory guided discriminative tracker for UAV object tracking" is the good contribution in the domain, and hot topic of research. Few suggstion must be incporporated in its revised verison before publication of the article: 

---Include one separate subsection in the introduction section: Objective, Motivation, and Contribution. Further, Scope of the study should clearly mention.

---Work flow should be represented in complete pseudorandom form must include input and output.

in section-2, Along with paragraphs some most recent relevant article (from year 2020 to 2022) should be done in tabular form by considering various parameters as needed. After Tabular study strong research gap must be created as currently it is very week, the following latest research article may be included in discussion:

Review for capacity and coverage improvement in aerially controlled heterogeneous network

Unmanned aerial vehicle-enabled layered architecture based solution for disaster management

EDTP: Energy and Delay Optimized Trajectory Planning for UAV-IoT Environment

Emerging UAV technology for disaster detection, mitigation, response, and preparedness

Proof read required for flow less English style and grammar. Further, conclusion and Future work should be strengthen…

Author Response

Thank you so much for your valuable and encouraging comments. We have adopted all your constructive suggestions to the current revised manuscript. Your comments and our one by one responses are listed in the attachment.

Reviewer 2 Report

The authors claim the implementation of a novel discriminative tracker called TMDiMP. Such a tracker is self-attention based using a novel memory-aware mechanism. Also, they have generated an object-tracking dataset named VIPUOTB.

Even though the topics could be interesting for the community, the paper lacks some aspects that must be corrected before the manuscript is ready to review.

So some non-exhaustive observations:

1.- Section 3 is irrelevant due to the following:

1a) It is too small

1b) It is just used to say that the DIMP algorithm exists. Then you can use a single phrase for it.

2.- Unfortunately, Section 4, where the proposed method is explained, misses the point. It is mainly because the explanation made of the created framework was not clear.

2a) It is recommended to explain in a better way all the subsections of the proposed framework shown in Fig. 2.

2b) It is highly recommended to include a simple example to clarify all the proposed algorithm.

3.- Sometimes is necessary to create a dataset, as in this case; however, section 5 is confusing. For example:

3a) The reason for including table 4 is unclear. Assuming that you are defining the dataset attributes: Why add a confusion matrix for the considering parameters? 

3b) the authors indeed mentioned that there is no benchmark dataset devoted enough to visual tracking. However, they must better justify the reason for creating a new one.

3c) Figures 4 and 5 are confusing too. It is imperative to explain in depth.

Besides, Figure 4 shows that the other datasets are better than the ones created.

Please, add more detailed information about them.

4.- Results lack consistency.

4a) Figures 15 and 2 are the same. Then, What is the finality of repeating it?

4b) Caption for Figure 15 says, "The architecture of the tracker using the classic attention method."  From this caption, it is waiting for the architecture for the proposed method.4c) Tables 5, 6, and 8 compare results for the proposed method and the own dataset,  right?Then, where are the results for the other datasets?4d) There are a lot of graphics where the comparison is presented, but there is not easy to see the difference between the methods or datasets used. Perhaps a table could be a better choice to show this information.

Author Response

Thank you so much for your valuable comments. We have adopted all your constructive suggestions to the current revised manuscript. Your comments and our one by one responses are listed in the attachment.

Reviewer 3 Report

TMDiMP: Temporal memory guided discriminative tracker for UAV object tracking  

This paper builds a UAV object tracking dataset named VIPUOTB and designed framework with end-to-end training capabilities, called TMDiMP, which embeds a novel memory-aware attention mechanism. In terms of experimental design, firstly, the experiment is compared with the existing UAV target tracking algorithm in multiple attributes, and the performance of the algorithm proposed in this paper is also very good. Then, the algorithm is compared on the existing three publicly available data sets. The proposed algorithm still performs well, and the workload of the whole experiment part and the comparison aspect are sufficient.

1.       In line 32, the text mentions that “Most of the objects in video sequences captured by UAV occupy less than 1% of the total number of pixels”. But the proportion of the objects in Figure 1 seems to be greater than 1%.

2.       The experimental results were discussed and analyzed in detail, and the relevant ablation experiments were also conducted later, but the ablation experiments can be introduced in more detail. The overall experimental design is good.

3.       There is little introduction to the algorithms compared in the paper, the description of the methods compared is not enough. The author should add a more detailed description of relevant comparison methods.

4.       Please explain the use of the confusion matrix in Table 4

5.       There are many spelling mistakes in the paper, and the English expression is colloquial, so it is necessary to make appropriate English expression changes. For example: Misspell UAV as UVA; Incorrect section reference at end of line 390, 397.

6.       Some tables and figures need to be optimized appropriately, such as Table1, Table3 and Figure 7

Author Response

(The authors gave the same response as above.)

Reviewer 4 Report

The article is very well written and almost error free.

Therefore, there is no need to send a file with changes.

Author Response

Thank you so much for your valuable and encouraging comments.

Round 2

Reviewer 1 Report

All the suggestion is incorporated in revised manuscript, now it may be accepted in its current form.

Author Response

Dear reviewer, thank you so much for your professional and encouraging comments.

Reviewer 2 Report

After revising this new version, I realize the amount of work done by the authors to improve the manuscript. Therefore, I think the paper is ready to be considered by the editors for publishing, even if minor details could be improved.

Author Response

(The authors gave the same response as above.)

Reviewer 3 Report

The paper solves the problem of small targets and camera motion on UAV target tracking, but needs to be supplemented with some background and application aspects of the proposed technical solutions related to the topic of this journal.

Author Response

Dear reviewer, thank you so much for your professional comments. We have supplement with some background and application aspects of the proposed technical solutions in Section 1.

Line 43 of the fifth paragraph in Section 1, page 2. “In fact, the small object and camera motion are two common attributes in the aviation remote sensing field. For example, urban traffic surveillance has the characteristics of high space complexity [15]. In order to obtain more information about traffic scenes, the altitudes of UAV views are usually high, which leads to small sizes of objects. Besides, it is necessary to adjust shooting angles of the onboard cameras, otherwise, the UAV will lose the tracked objects that move very fast.

[15] Xu, H.; Cao, Y.; Lu, Q.; Yang, Q. Performance Comparison of Small Object Detection Algorithms of UAV based Aerial Images. In Proceedings of the 2020 19th International Symposium on Distributed Computing and Applications for Business Engineering and Science (DCABES), 2020, pp. 16–19. https://doi.org/10.1109/DCABES50732.2020.00014.